# Novel Soft-Switching Integrated Boost DC-DC Converter for PV Power System

**Khairy Sayed** [1] , **Mohammed G. Gronfula** [2] **and Hamdy A. Ziedan** [3],*

1   Faculty of Engineering, Sohag University, Sohag 82524, Egypt; Khairy.f@gmail.com
2   Faculty of Engineering, Alasala Colleges, Dammam 31483, Saudi Arabia;
    mohammed.gronfula@alasala.edu.sa
3   Faculty of Engineering, Assiut University, Assiut 71518, Egypt
*   Correspondence: ziedan@aun.edu.eg; Tel.: +20-10-2704-4488

**Abstract:** This paper presents a novel soft-switching boost DC-DC converter, which uses an edge-resonant switch capacitor based on the pulse width modulation PWM technique. These converters have high gain voltage due to coupled inductors, which work as a transformer, while the boost converter works as a resonant inductor. Upon turning on, the studied soft switching circuit works at zero-current soft switching (ZCS), and upon turning off, it works at zero-voltage soft switching (ZVS) while using active semiconductor switches. High efficiency and low losses are obtained while using soft switching and auxiliary edge resonance to get a high step-up voltage ratio. A prototype model is implemented in the Power Electronics Laboratory, Assiut University, Egypt. Seventy-two-panel PV modules of 250 W each were used to simulate and execute the setup to examine the proposed boost converter.

**Keywords:** novel soft switching; boost DC-DC converter; ZCS; ZVS; auxiliary edge resonant; switch capacitor; PV power system

## 1. Introduction

Recently, vigorous attempts have been made to attain optimized switch-mode power converter circuits with which to interface renewable energy sources to a utility such as a solar photovoltaic PV, fuel cell, and supercapacitor banks. To increase the dependence of the power electronics system and the power conversion capability, the expansion of boost DC-DC circuits is needed with modern control schemes [1–5]. Solar photovoltaics have been introduced as a future power source, due to the increasing need for clean energies and power-distributed generation systems. Injecting the generated energy into the network (grid) requires power conditioning circuits. In general, a higher step-up ratio DC-DC circuit is required for boosting the low PV voltage to a higher voltage output. Using conventional circuits such as cascade DC-DC power converters leads to additional cost and system complexity. Moreover, the conventional fly-back DC-DC converter topologies have leakage constituents that lead to voltage stresses on the semiconductor switches and more power losses, resulting in lower circuit efficiency.

To improve the energy conversion efficiency of the PV system in this paper, a soft-switched boost-type converter using an additional simple resonant cell is adopted. The auxiliary resonant circuit consists of a resonant inductor, an auxiliary switch, auxiliary diodes, and a resonant capacitor. As mentioned above, the traditional boost converter has decreased efficiency due to regularly being turned on and off, which generates switching losses. As we know, increasing the switching frequency will increase the periodic switching and conduction losses, resulting in an increase in the energy loss of the total system. To decrease these losses, we propose a soft-switching arrangement. This can be

achieved by adding an auxiliary circuit, as shown in Figure 1a, instead of a classical hard-switched converter [6]. Moreover, the auxiliary circuit has no additional complexity or cost.

To reduce the weight and size, a higher switching frequency operation, i.e., in the range of more than 50 kHz, is utilized [7]. In traditional switching converters, the exemplary switching frequency is between 5 kHz and 20 kHz. The switching frequency generally cannot be increased because this results in increasing switching losses and stresses on semiconductor devices. A high frequency is utilized to reduce the volume and weight of passive devices. The dynamic performance is better in the case of higher frequencies. High-frequency switching speeds up the converter response time and reduces the output filter volume, cost, and size. A high switching frequency is desirable for the minimum output inductor size and maximum control loop bandwidth. Overall system size will be reduced due to the operation of switches at a high frequency, which will make the new converters more feasible. To overcome these problems, soft-switching schemes such as zero-voltage soft switching (ZVS) type or zero-current soft switching (ZCS) commutation should be utilized to reduce switching losses and the semiconductor device stresses. For power converters in a distributed PV system, a soft-switching technique is proposed [8–12]. The presented converter (see Figure 1b) has advantages of improved efficiency at a higher switching frequency, low leakage current, wide load range, reduced weight and size, and a maximum total efficiency of 97.1% at a switching frequency of 100 kHz.

The problems of excessive electromagnetic interference EMI and low efficiency can be solved by using zero-voltage-transition (ZVT) converters by limiting the turning-off $di/dt$ in the output-side rectifier. Various types of ZVT converters have been presented before [9,13], but these converters suffer from several drawbacks, such as the following.

(1) Most ZVT converter topologies have the auxiliary switch with hard-switching turned off, which, of course, limits the efficiency of such converters; see Figure 1c [13].
(2) The auxiliary circuits increase the complexity of the system because they consist of several active and passive components. Moreover, the auxiliary semiconductor switches require a floating gate drive.
(3) The elements of auxiliary circuits suffer from a higher voltage and current stresses.
(4) The conduction losses are high in auxiliary circuit components [14].

Figure 1d shows the configuration circuit of boost type based on an edge-resonance ZVS-PWM DC-DC converter circuit utilizing IGBTs as active switches [15]. This converter is considered a conventional chopper-fed, boost-type power converter circuit. This converter also includes an auxiliary active resonance-snubber circuit, which includes a resonance capacitor $C_r$, a resonant inductor $L_r$, an auxiliary active power switch $S_2$, a lossless snubber capacitor $C_s$, and diode $D_2$ in the auxiliary circuit.

An improved circuit of a DC-DC power converter was proposed in [16]. The problem in this power electronics circuit, shown in Figure 1e, is that the auxiliary switch is turned-off under the hard-switching condition. However, the used auxiliary switch in the chopper-fed DC-DC circuit, shown in Figure 1e, can operate under the ZCS operation condition at turn-off transition by adding a simple resonance capacitor $C_r$.

Recent research has been based on the combination of a PV module with a power converter, which addresses all the required control requirements, such as the tracking of maximum and output voltage regulation. An inclusive literature review for nonisolated single-phase PWM inverters for a PV-integrated AC module is presented in [17,18].

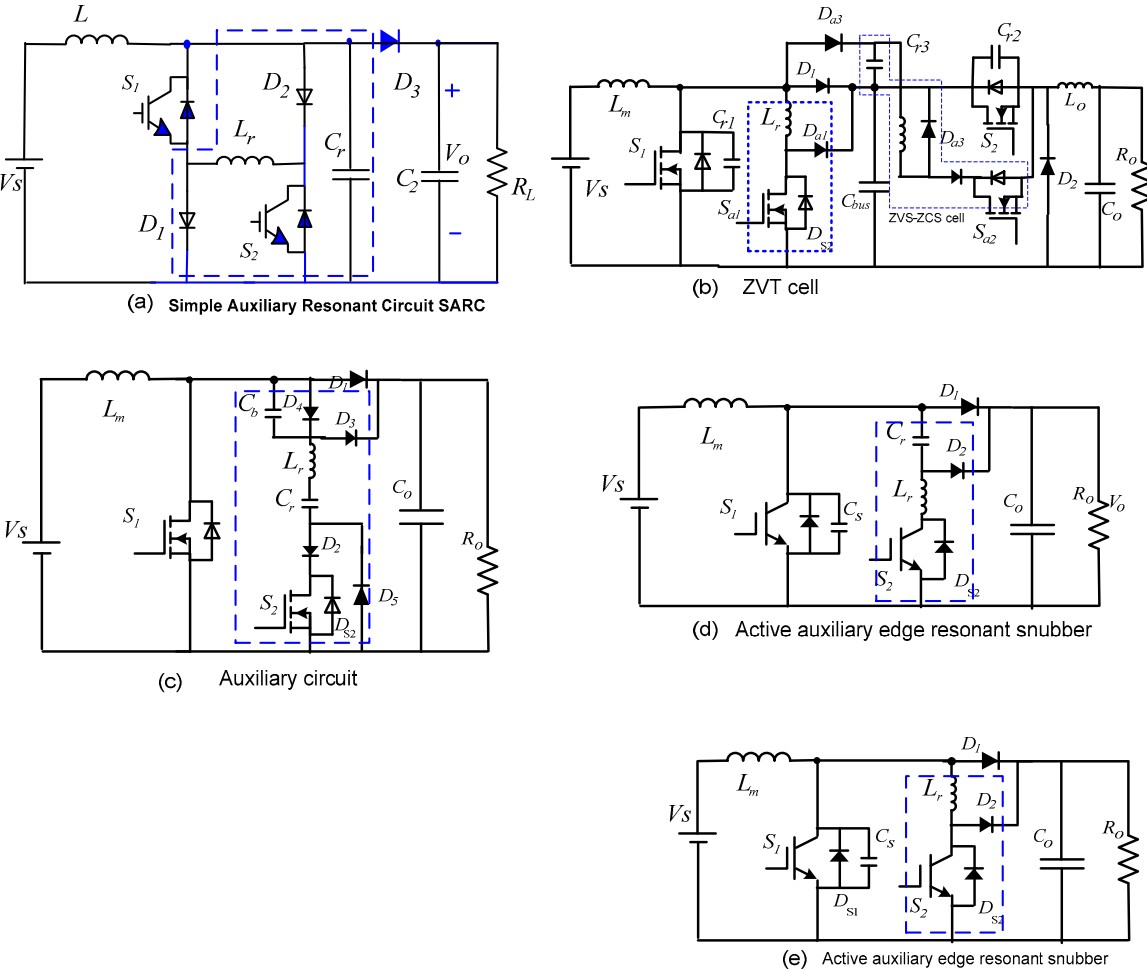

**Figure 1.** (**a**) Circuit diagram of the soft-switched boost converter proposed in [6]. (**b**) The soft-switching PV inverter proposed in [7]. (**c**) Zero-voltage-transition (ZVT) circuit proposed in [13]. (**d**) Zero-voltage soft-switching boost converter with auxiliary edge-resonance snubbers proposed in [15]. (**e**) Boost PWM ZVT circuit proposed in [16].

In [19], parallel-connected converters for solar PV arrays are studied. Similar to the AC solar PV arrays applications, the output voltage of solar PV arrays is low, so a high-efficiency DC-DC converter is required. The high gain nonisolated DC-DC circuits which are used in the implementation of converter circuits are studied [20–23]. Previous works have used soft-switching coupled inductors [24–36].

A novel circuit for the boost DC-DC converter interfacing PV systems is introduced in this paper. In the following section, the paradigm design details of the suggested circuit are revealed. The behavior of the inverter system stage was similar to constant power load, not like resistive load, by placing a battery load at the output of the boost stage. The boost converter stage can be managed to draw a certain amount of current determined by the inductor rating and the duty cycle, depending on the loading conditions. The introduced converter is also studied in continuous and discontinuous modes of conduction. Also, the efficiency of the converting system is improved. The converting system consists of a solar PV array and a soft-switching boost DC-DC converter, which uses an auxiliary resonant circuit, auxiliary switch, diode, inductance, and capacitance. The traditional boost converters have lower efficiency due to hard switching, which increases losses due to on/off operations.

In Section 2, PSIM is presented, which is a well-known simulation program assisting in the modeling and simulation of the solar PV system and consists of a closed-loop control converter system. The strategy of control and steady-state analysis is discussed. Section 3 clarifies the operation principle via the operation interval waveforms and the relevant equivalent circuits. Section 4 presents the

control scheme of the proposed power converter. The experimental results of the implemented boost converter are presented in Section 5. Using several power levels, output voltages and currents are studied and measured.

## 2. Design of the Circuit

The edge-resonance switched-capacitor (ER-SC) converter includes two active IGBT switches, i.e., $S_1$, $S_2$, connected to two auxiliary diodes $D_1$, $D_2$. The resonance is formed by a resonance capacitor $C_r$ and a resonance coupled inductor $L_r$. The circuit configuration of the proposed soft-switching ER-SC boost DC-DC converter circuit is illustrated in Figure 2. A voltage doubler is composed of $D_4$, $C_2$, and coupled inductor $L_r$. The main advantages of the proposed soft-switching ER-SC boost PWM DC-DC converter are as follows.

(1) Higher step-up voltage gain can be obtained due to the occurrence of edge resonance formed by elements $L_r$ and $C_r$.
(2) Power losses are reduced due to soft-switching operation; this improves efficiency.
(3) ER-SC involves a module or modules as the basis of a configurable design by simple circuit construction and H-bridge IGBT module.
(4) The current participation operation is obtainable between $S_1$ and $S_2$, which is efficient for the applications that have high current input.
(5) The PWM gate signal for $S_1$ and $S_2$ can be general; thereafter, the gate driver circuits are simpler than that for the conventional converter.

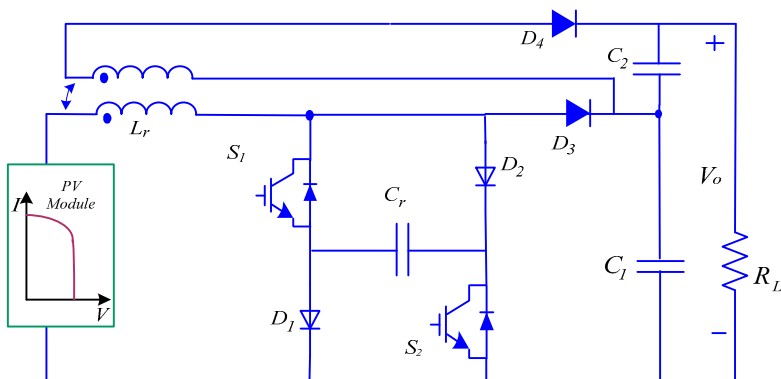

**Figure 2.** Proposed soft-switched ER-SC boost DC-DC converter.

## 3. Principle of Operation

During each commutation period of the boost converter operation, there are five operating intervals. The operating current waveforms are illustrated in Figure 3. Five switching intervals take place through one operating period. The subsequent equivalence circuits of operating intervals of the studied circuit during one switching period are shown in Figure 4. To analyze the proposed DC-DC converter circuit, the output current and voltage waveforms are shown in Figure 5. The theory of operation of the converter is demonstrated in the subsequent sections using the corresponding switching interval equipollent circuits.

### 3.1. Interval 1: $t_o \leq t < t_1$ ($S_1$ and $S_2$: ON, $D_1$, $D_2$, and $D_3$: OFF)

This mode is called the ZCS turn-on interval. The active power switches $S_1$ and $S_2$ turn on simultaneously at time $t_0$. Then, the current passing through coupled inductor $i_{Lr}$ and the semiconductor IGBTs currents $i_{S1}$ and $i_{S2}$ increases progressively from an initial value of zero with edge resonance caused by the use of $L_r$ and $C_r$. Consequently, $S_1$ and $S_2$ can achieve ZCS turn-on commutation.

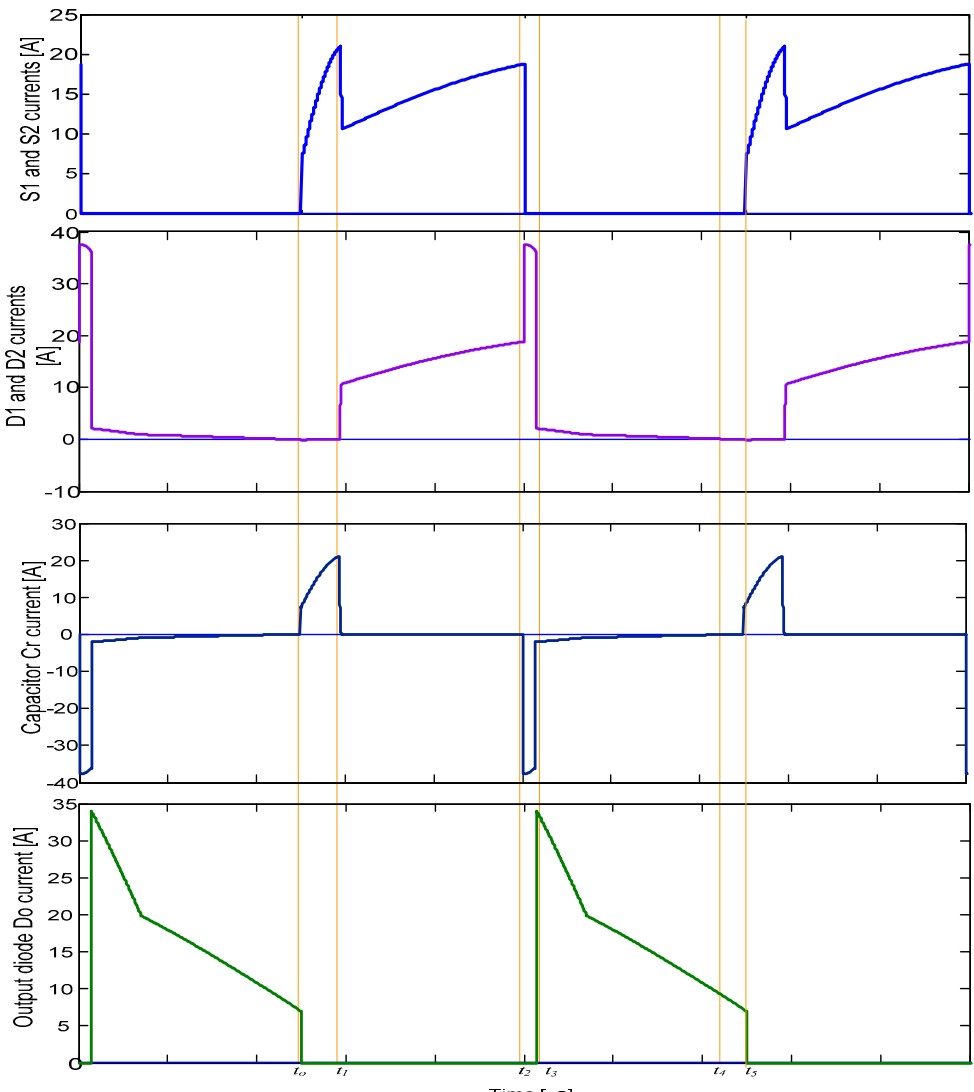

**Figure 3.** Operating waveforms.

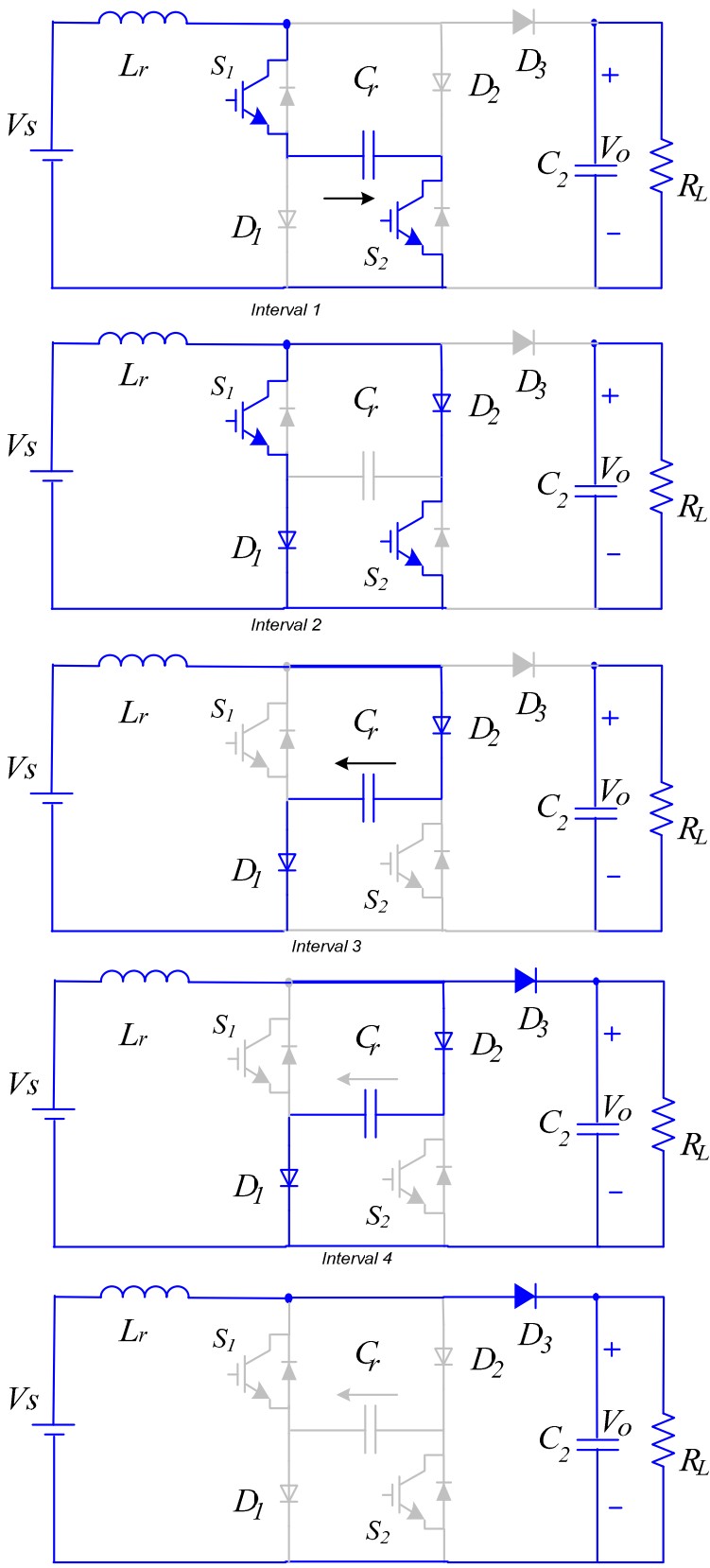

**Figure 4.** Equivalent circuits of operation intervals.

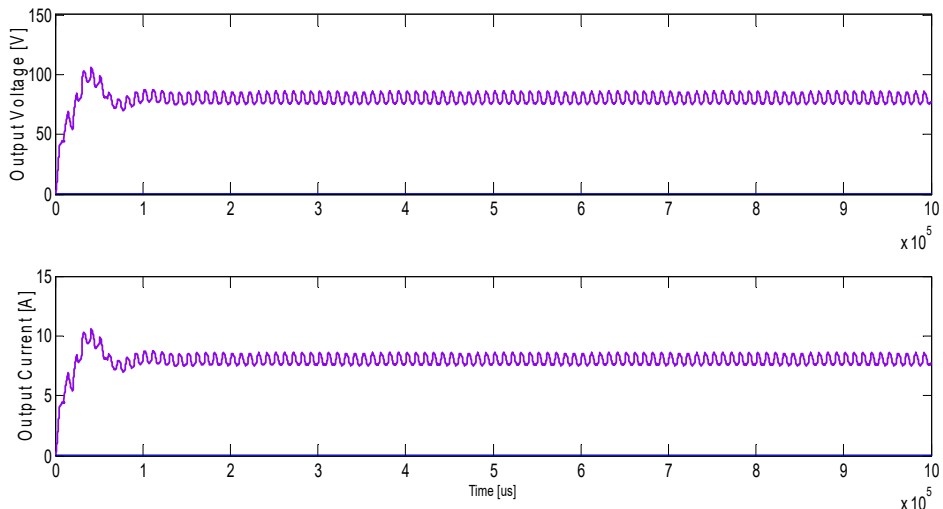

**Figure 5.** Voltage and current output of the proposed DC-DC converter at a 0.5 duty cycle.

The resonance capacitor $C_r$ has been discharged in this interval by the edge-resonant current flow in the ER-SC circuit, as shown in Figure 3. During this interval, the equivalent circuit during resonance operation can be simplified, as shown in Figure 6. *Req* is the equivalent series resistance of the circuit; it is a simple RC circuit. The resonant inductor current and resonant capacitor voltage can be obtained using the Laplace transform.

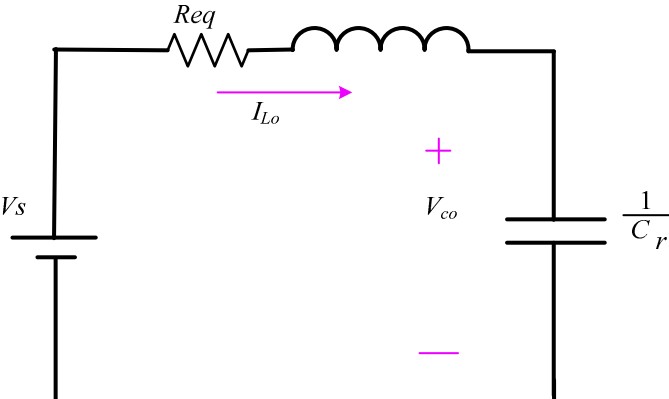

**Figure 6.** Equivalent S-domain of the resonant circuit.

The transfer function for the loop current $I_{Lr}$ is

$$H_I(s) = \frac{I_{Lr}}{V_S} = \frac{1}{R + SL + (SC)^{-1}} = \frac{SC}{S^2LC + SRC + 1} \tag{1}$$

The transfer function for the loop current $V_{cr}$ is

$$H_I(s) = \frac{V_{Lr}}{V_S} = \frac{(SC)^{-1}}{R + SL + (SC)^{-1}} = \frac{SC}{S^2LC + SRC + 1} \tag{2}$$

By neglecting the equivalent resistance, the characteristic equations can be solved to get $i_{Lr}$ and $v_{cr}$. Then, the current $i_{Lr}$ can be expressed as

$$i_{Lr} = \frac{V_{in} + V_o}{Z_r} \sin \omega_r (t - t_o) \tag{3}$$

where $Z_r = \sqrt{L_r/C_r}$ and $\omega_r = \sqrt{L_r C_r}$.

The current flow in the resonantly coupled inductor $i_{Lr}$ at time $t = t_1$ is obtained by the following relation:

$$I_{Lr1} = i_{Lr}(t_1) = \frac{\sqrt{V_o(2V_{in} + V_o)}}{Z_r} \tag{4}$$

$$t_1 = \frac{1}{\omega_r} \cos^{-1} \frac{V_{in}}{V_{in} + V_o} \tag{5}$$

The voltage across the resonant capacitor can be expressed as

$$v_{Cr}(t) = V_{Co} \cos \omega_r(t) \tag{6}$$

### 3.2. Interval 2: ($S_1$ and $S_2$: ON, $D_1$, and $D_2$: ON and $D_3$: OFF); $t_1 \leq t < t_2$

This mode is called inductive energy storage interval because storing energy is in coupled inductance in this interval. The edge-resonance capacitance $C_r$ is discharged at time $t_1$; at this moment, the current flows through forward-biased diodes ($D_1$ and $D_2$). The current in coupled inductor $I_{Lr}$ increases linearly during this interval and is expressed by

$$i_{Lr} = \frac{V_{in}}{L_r}(t - t_1) + I_{Lr1} \tag{7}$$

The current through coupled inductor $i_{Lr}$ is equally divided into two branches, $S_1$–$D_2$ and $S_2$–$D_1$. From Equation (7), $i_{Lr}$ at time $t_2$ can be calculated by

$$I_{Lr2} = i_{Lr}(t_2) = \frac{V_{in}}{L_r}(DT - t_1) + I_{Lr1} \tag{8}$$

where $D$ indicates the duty ratio of switches $S_1$ and $S_2$, which may be described by

$$D = T_{on}/T \tag{9}$$

### 3.3. Interval 3: ($S_1$ and $S_2$: OFF, $D_2$, and $D_1$: ON and $D_3$: OFF); $t_2 \leq t < t_3$

Both the main IGBT power devices $S_1$ and $S_2$ are commutated (turned-off) simultaneously at time $t_2$ under the ZVS condition. The capacitor $C_r$ is completely charged by edge resonance, while the voltages on terminals of $S_1$ and $S_2$ increase gradually due to the influence of capacitor $C_r$. Therefore, soft-switching ZVS turn-off can be realized in switches $S_1$ and $S_2$. The current $i_{Lr}$ throughout this interval is expressed as

$$i_{Lr} = I_{\max} \sin \left\{ \omega_r(t - DT) + \tan^{-1} \frac{Z_r I_{Lr2}}{V_{in}} \right\} \tag{10}$$

where $I_{max}$ denotes the maximum value of $i_{Lr}$, which can be defined by

$$I_{\max} = \sqrt{I_{Lr2}^2 + \left( \frac{V_{in}}{Z_r} \right)^2} \tag{11}$$

This interval continues in operation until the voltage over resonant capacitor $v_{Cr}$ equalizes the output voltage $V_o$ at time $t_3$, and the subsequent current through an inductor can be determined by

$$I_{Lr3} = i_{Lr}(t_3) = \sqrt{I_{\max}^2 - \left( \frac{V_o - V_{in}}{Z_r} \right)^2} \tag{12}$$

$$t_3 = \frac{1}{\omega_r} \left( \sin^{-1} \frac{V_o - V_{in}}{Z_r I_{\max}} + \tan^{-1} \frac{V_{in}}{Z_r I_{Lr2}} \right) + DT \tag{13}$$

*3.4. Interval 4: ($S_1$ and $S_2$: OFF, $D_2$, and $D_1$: ON and $D_3$: ON); ($t_3 \leq t \leq t_4$)*

This interval is called the inductor energy release interval because the stored energy is allowed to escape through $D_3$. In this interval, the resonant voltage across capacitor $v_{Cr}$ increases gradually up to the value of the output voltage at time $t_4$, then the conduction period of diodes $D_1$ and $D_2$ completes at the end of this interval. However, the current passing through inductor $i_{Lr}$ is delivered to the load through $D_3$, and therefore, the input voltage $V_{in}$ is stepped-up to the output voltage $V_o$. Thereby, the occurrence of the reverse recovering current for the output freewheeling diode $D_3$ can be alleviated.

*3.5. Interval 5: ($S_1$ and $S_2$: OFF, $D_1$ $D_2$: OFF and $D_3$: ON); ($t4 \leq t \leq t5$)*

At $t_4$, diodes $D_1$ and $D_2$ stop conducting and turn off at ZVS. Inductor current $i_{Lr}$ reduces to its minimum value at $t_5$. The load current flows through the output diode $D_3$.

## 4. Analysis of Gain Voltage of the Converter

In the *First Mode*, switches $S_1$ and $S_2$ are conducting. The average voltage across the magnetizing inductance $L_m$ and capacitors $C_2$ and $C_1$ is given as

$$V_{Lr}^{ON} = V_{in} \tag{14}$$

where $N$ is the coupled inductor turns ratio

$$V_{C2} = NV_{Lr}^{ON} \tag{15}$$

In the *Second Mode*, switch $S_1$ and $S_2$ are in the OFF mode, which can be expressed as

$$V_{Lr}^{OFF} = V_{in} - V_{C1} \tag{16}$$

$$V_{C2} = -NV_{Lr}^{OFF} \tag{17}$$

From the magnetizing inductance, volt-second balance can be expressed as

$$V_{C1} = \frac{V_{in}}{1 - D} \tag{18}$$

From Equations (14) to (18), one obtains

$$V_{C2} = \frac{NV_{in}}{1 - D} \tag{19}$$

The output voltage of the converter is the sum of average voltage across capacitors $C_1$ and $C_2$. Therefore,

$$V_O = V_{C1} + V_{C2} \tag{20}$$

Hence, from Equations (19) and (20), the gain of the ideal voltage of the proposed boost converter is obtained as

$$M = \frac{V_O}{V_{in}} = \frac{1 + N}{1 - D} \tag{21}$$

From Equation (21), it is clear that while increasing the turns' ratio of the coupled inductor, voltage stress on the semiconductor switches decreases, and the voltage gain of the proposed converter increases.

## 5. Circuit of Control

An approach of Perturb and observe (P&O) is used to track the maximum power point (MPPT) of the array, which is suitable for implementation in a PIC microcontroller. This process is repeated to track changes in solar radiation level and temperature. The control technique of MPPT is obtained in

the block of logic and generates a base voltage to regulate the duty cycle to move the point of operation on the P-V curve of the PV module. The reference voltage is compared with the output voltage using a comparator. Figure 2 shows that switches $S_1$ and $S_2$ are operated due to the pulse PWM switching signal. The output voltage is controlled within particular limits.

Figure 7 shows a block diagram of the control system. The implemented converter is controlled by the PIC microcontroller, which is used to develop the P&O MPPT algorithm. In this case, the outmost feedback loops have to be connected to the output terminal of the associated power electronic converters, and hence, the input terminal will behave as a negative incremental resistor at low frequencies [37]. This property will limit the operation of the PV interfacing converter in either the constant voltage or the constant current region of the PV generator to ensure stable operation. The boost DC-DC converter can be applied as a voltage- or current-fed converter, limiting the stable operation region accordingly [37].

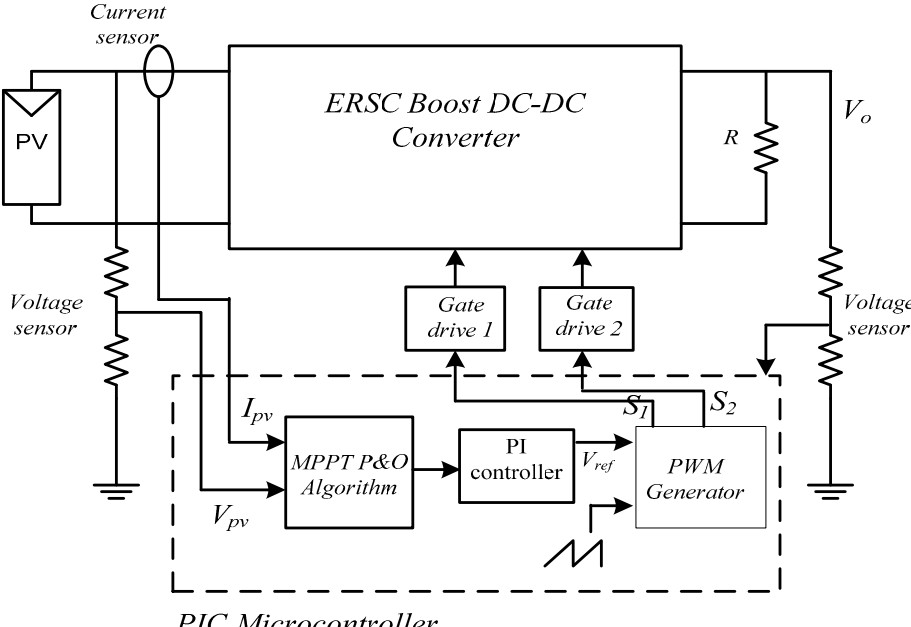

**Figure 7.** Soft-switching DC-DC boost converter block diagram.

The PV models work as a voltage source, which is not true under these conditions. The PV panel is a highly nonlinear input source with two distinct source regions [38]. Its low-frequency dynamic output impedance (i.e., incremental resistance) behaves similarly to interfacing converters. At the MPPs, the PV-generator dynamic and static resistances are equal [38]. The dynamic changes in the PV panel interfacing converter are caused by the operating point-dependent dynamic resistance, which is very high and equal to static resistance. The PV panel behavior and its effects on the dynamics of the associated interfacing converters have been studied to avoid problems in designing the interfacing converters and the related energy systems for PV applications [38].

## 6. Experimental Results

The studied soft-switching circuit was implemented as a prototype to demonstrate the operation principle. The input voltage is taken from the PV module, monocrystalline, 250 W, $V_{mp}$ = 30.7 V, $I_{mp}$ (max. power current) 8.15. A 600 V-40 A IGBT was used as a switch operating at a switching frequency of 40 kHz, and the duty cycle was changed to prove the operation technique. The gate signal was produced utilizing the PIC microcontroller, and thereafter, the PWM signal was delivered to an optocoupler TLP250 to drive the IGBT switch. Optical isolation was the major advantage of using this driver compared to other drivers. TLP 250 was used to implement fast switching of power devices and

reduce associated switching power losses. TLP 250 as an optocoupler provides a very high isolation voltage. It can easily be interfaced with a microcontroller. The circuit parameters are listed in Table 1.

**Table 1.** PV module prototype parameters.

| Parameter | Value, Unit |
|---|---|
| Rated output power $P_o$ | 500 W |
| Input voltage $V_{in}$ | 30 V |
| Output voltage $V_o$ | 100 V |
| Boost inductor $L_r = L_1 = L_2$ | 900 μH |
| Mutual inductance $M$ | 770 μH |
| Resonant capacitor $C_r$ | 60 nF |
| Variable load resistor $R_o$ | 20 Ohm |
| Output smoothing capacitor $C_1 = C_2$ | 500 μF |
| Switching frequency | 40 kHz |
| Photovoltaic Module | Monocrystalline, 250.0 Watt, $V_{mp}$ = 30.7 V, $I_{mp}$ = 8.15 A |

Figures 8 and 9 show the waveforms of output voltage and current of two switches under different loading conditions. As shown, both the switches can turn on and turn off with zero-current soft switching. Subsequently, a reduction in the switching losses can be obtained, and an improvement in the efficiency of the converter can be achieved. Figure 10 shows the output voltage waveform at standard irradiance, i.e., 1000 W/m$^2$, and full-load condition.

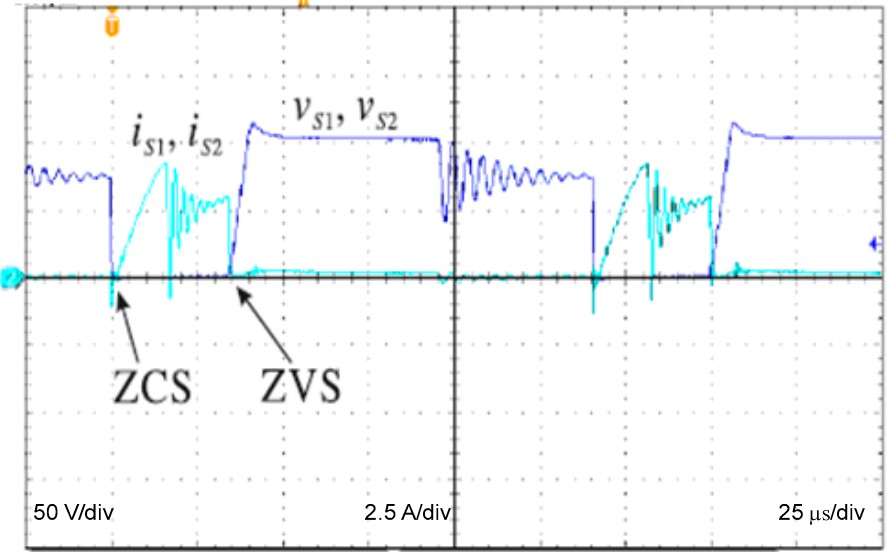

**Figure 8.** Waveforms of voltage and current of switches $S_1$ and $S_2$ at Po = 200 W.

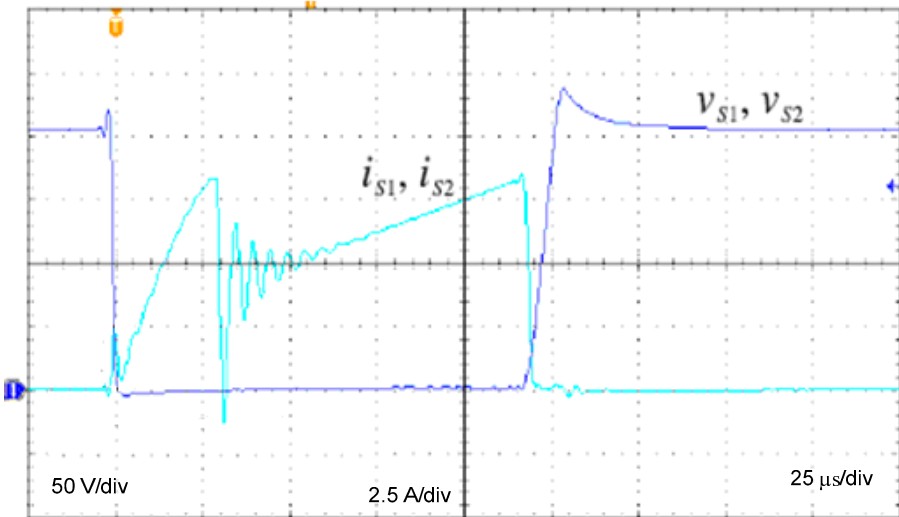

**Figure 9.** Operating waveforms of voltage and current of switches $S_1$ and $S_2$ at $P_o$ = 500 W.

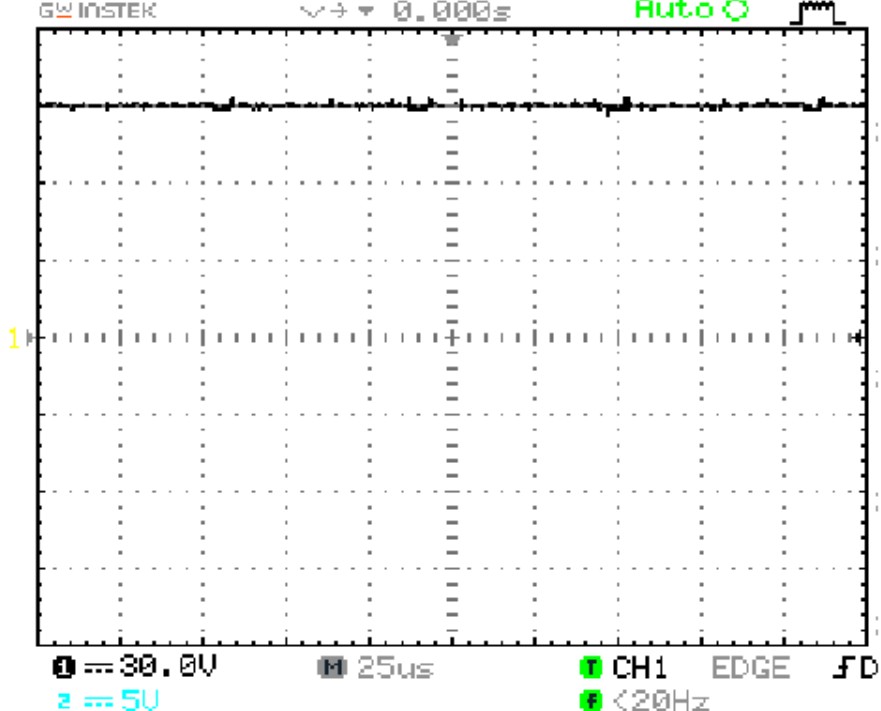

**Figure 10.** Output voltage at standard irradiance and full-load condition.

Figure 11 shows the measured efficiency of the proposed boost DC-DC converter. The developed circuit has better efficiency than the previously-developed [6] hard-switching converter. However, the maximum obtained efficiency is 97.1% at full load, i.e., 500 W; about a 3% improvement in efficiency is gained. The switching losses are about 5 W at a switching frequency 40 kHz. A photograph of the experimental setup hardware is shown in Figure 12. The PV array installation on the roof is shown in Figure 13.

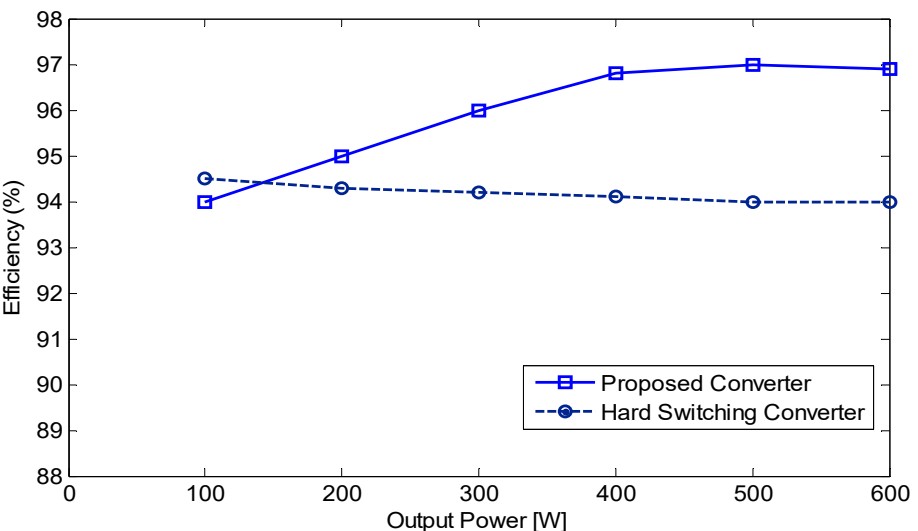

**Figure 11.** Efficiency which measured for the prototype DC-DC converter.

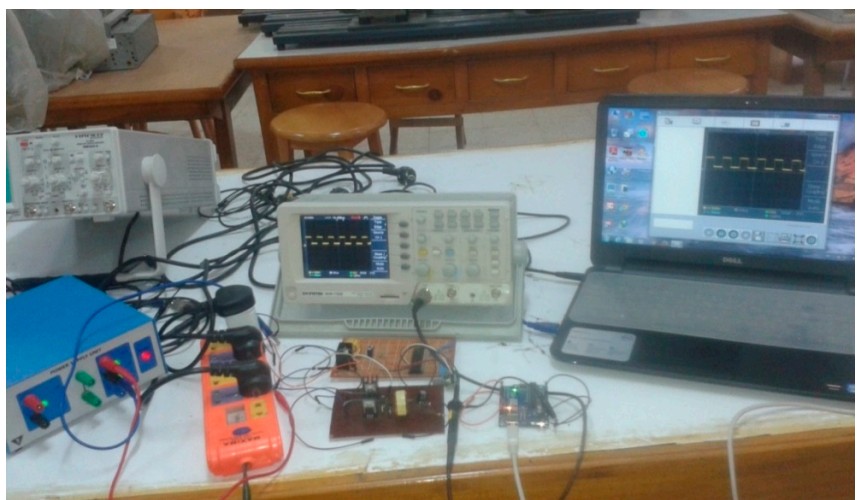

**Figure 12.** Hardware installation of the experimental setup.

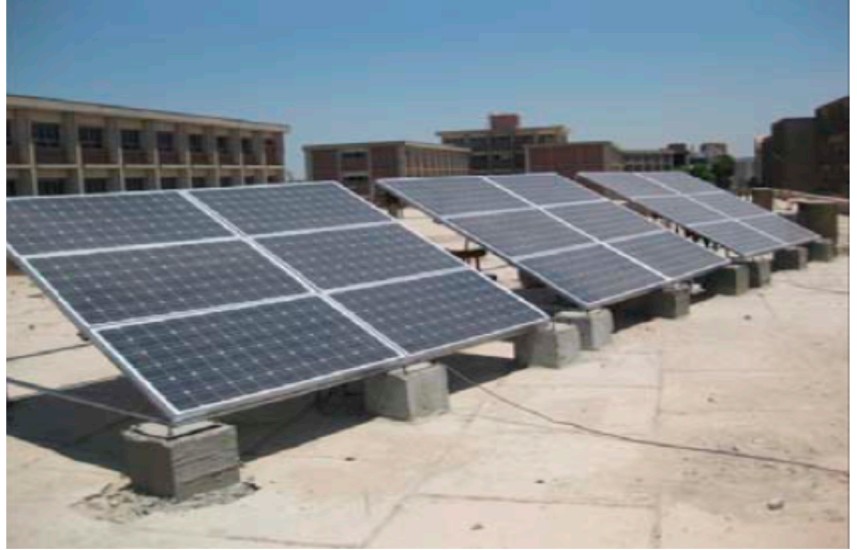

**Figure 13.** The PV array installation on the roof (250-W, 72-cell PV module)

## 7. Power Loss Analyses

The section presents a comparative analysis of previously-developed hard-switched and the proposed soft-switched converters. Analytical calculations are made to calculate the losses of components, voltage, power, and frequency of switching, and the increase in temperature due to the use of semiconductors devices. Figure 14 shows the power losses as a function of the input current. The compared power converters are realized and investigated experimentally. Figure 15 is used to analyze the dissipated losses in the two converters and each of their passive and active components. The power loss breakdown of the key components of the implemented converter is estimated. Figure 16 shows the losses breakdown (calculated value) at half- and full-loading condition of the proposed circuit.

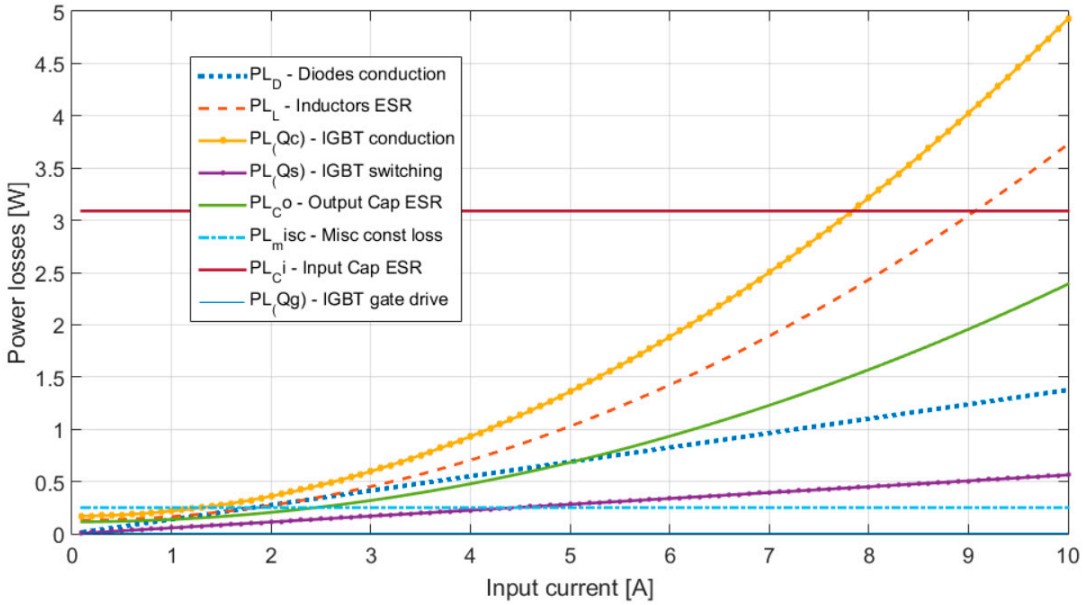

**Figure 14.** Calculated losses in each component of the proposed PV converter.

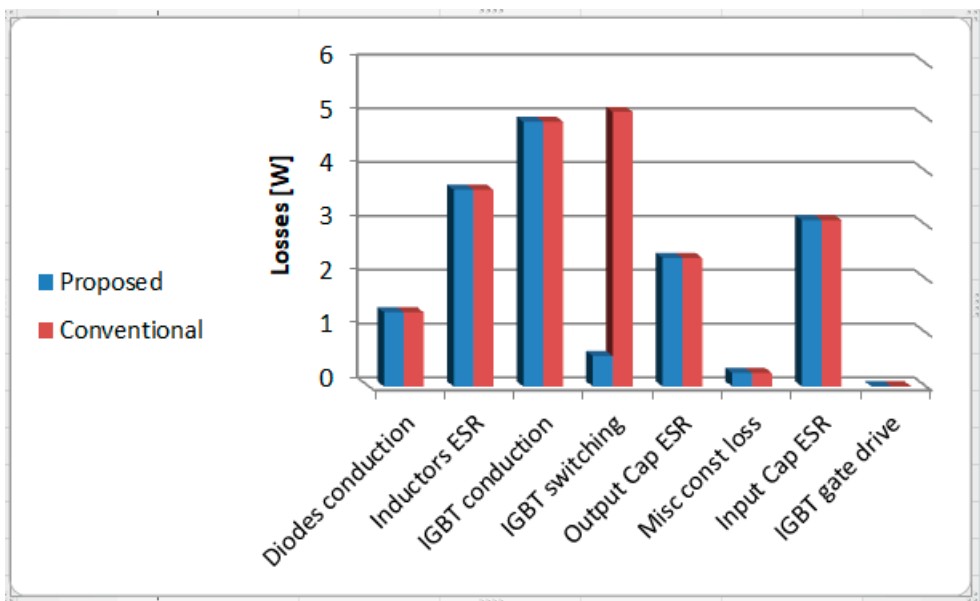

**Figure 15.** Losses comparison between the hard-switched and proposed converter.

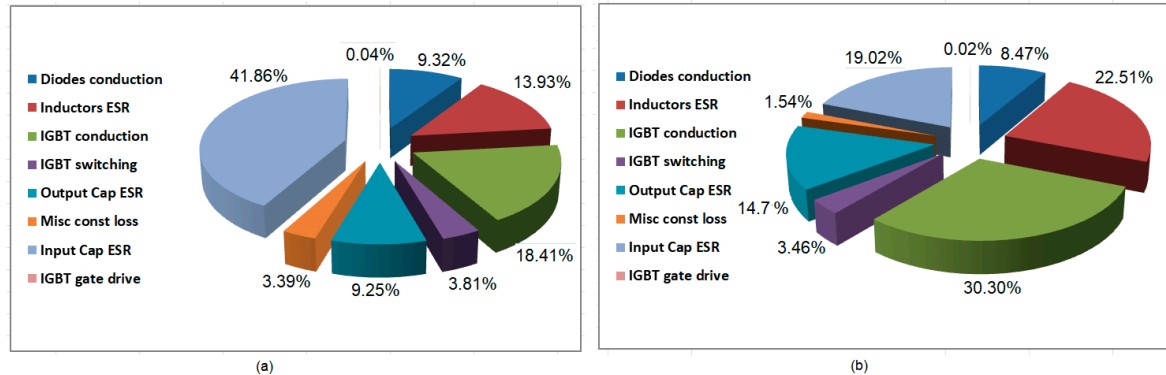

**Figure 16.** Breakdown losses. (**a**) Using half-load (Po = 250 W); and (**b**) Using full-load (Po = 500 W).

Table 2 shows a comparison between the proposed circuit with different topologies used with PV modules. From this table, the proposed topology has a lower switching frequency and a higher efficiency.

**Table 2.** Comparison with other existing topologies.

| Parameter | Proposed | Ref. [6] | Ref. [7] | Ref. [8] | Ref. [3] |
|---|---|---|---|---|---|
| Input voltage | 30 | 150–230 | 100–200 V | 90–256 | 23 |
| Output voltage | 100 | 380 | 300 | 400 | 30 |
| Voltage gain | 3 | 2.53 | 1.5–3 | 1–4 | 1.5 |
| Switching frequency | 40 k | 30 k | 100 k | 100 kHz | 100 k |
| Rated power | 500 W | 700 W | 1000 | 250 | 10–60 W |
| Efficiency | 97 | 96 | 97 | 95 | 92–94% |

## 8. Conclusions

A novel circuit with a soft-switching, high voltage gain boost DC-DC converter was investigated for PV module integration. The operation of the PV panels is analyzed to study the equivalent circuit of each model and its operation. The coupled inductor can be used to maximize the voltage gain of the proposed nonisolated boost PWM DC-DC converter according to the winding turns ratio. The control technique is discussed for the proposed ER-SC boost-type soft-switching DC-DC power converter. A simulation model of the PV system was developed using PSIM to validate the developed converter. Then, the results of the experimental work are introduced using a prototype converter. Losses and the efficiency of the prototype converter were measured.

**Author Contributions:** Conceptualization, H.A.Z. and K.S.; methodology, H.A.Z.; software, K.S.; validation, M.G.G., and H.A.Z.; formal analysis, K.S.; investigation, H.A.Z.; resources, K.S.; data curation, K.S.; writing—original draft preparation, M.G.G., and H.A.Z.; writing—review and editing, K.S.; visualization, H.A.Z.; supervision, K.S.; project administration, H.A.Z.; funding acquisition, M.G.G. All authors have read and agreed to the published version of the manuscript.

**Funding:** This research received no external funding.

**Conflicts of Interest:** The authors declare no conflict of interest.

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
