# Peer review of "Novel Soft-Switching Integrated Boost DC-DC Converter for PV Power System"

_energies, doi:10.3390/en13030749_

Round 1
Reviewer 1 Report
This work presents a Novel Soft-Switching Integrated Boost DC-DC Converter for PV Power System.
I suggest a major revision of the work. In addition, I present some suggestions for the authors that could help to improve the quality of the paper.
1) Please improve the quality of the figures. Some of them are small and others are too big (fig. 3).
2) In the experimental prototype (Table I) the output voltage is 100V. However, in PV applications at least a DC=180V is required for a Grid connection V=110VAC. Please validate the topology with values for standard applications, or justify the reasons behind the selection of the design values.
3) Please explain about the efficiency analysis. Please clarify the reason about the efficiency decrease when output power is lower than 100W. Why it is lower compared with a hard switching converter (Fig. 10)?
Reviewer 2 Report
in this paper, a new DC-DC converter topology for PV systems is proposed resulting in lower losses and higher efficiency. Although the problem of high frequency operation with low losses is important, more explanations are needed for this work. More specifically:
The proposed implementation is based on a transformer system. However, many implementations nowadays avoid to use transformers due to extra losses and extra cost. In Section 2, the authors present the advances of their work. All these advances should be presented and highlighted within the paper. For example, the reduction of power losses are not included in Section 2. The simpler gate driver circuits and the simple circuit construction should be compared with other existing topologies. Furthermore, the novelty of this work should be more highlighted and proved, since it is used in the title as well. In Section 3: Principle of Operation, the authors should explain with more details how these equations are produced. It is very difficult to understand the connection among them. For example, how are equations (1), (2) and (3) produced? It would be better to draw the current in Figure 4. In experimental results, since 3 modules of 250W each are used, why is the rated output power equal to 500W and not 750W? A comparison with simulation results with other topologies showing voltages, currents, etc. would be very helpful. The use of English language should be improved. There are grammar and syntactical errors, e.g. line 10: "this paper is aimed at investigated", line 47: the auxiliary circuit not adds", line 50: the switching frequency could be increased hardly", etc.
Reviewer 3 Report
On p.1, line 19 in the abstract section, this reviewer cannot find the “250-W, 72-cell PV module” set up throughout the manuscript.
On p.2, line 46, Please show a figure of the classical hard-switched converter with a note to compare the differences between two types of converters.
On p.2, line 48, at line 43 in p.2, you mentioned that raising the switching frequency results in increasing periodic switching and conduction losses. Then, why are you trying to use higher frequency more than a classical converter?
On p.2, line 54, please show related literature reviews (more than 5) regarding the soft-switching technique applied to PV modules and point out what are the main contributions compare to those.
On p.2, line 55, without a comparison table compared to classical converters, this insists cannot be made since there are no references.
Round 2
Reviewer 1 Report
This work presents a Novel Soft-Switching Integrated Boost DC-DC Converter for PV Power System. The authors have well addressed the comments made by the reviewers. I suggest the publication of the work.
Author Response
I'd like to thank the reviewers for their valuable comments which enhance the quality of the paper.
Reviewer 2 Report
The authors did not address all the raised issues in the previous review. Their responses are very short, without specific explanations per issue. More specifically:
It is fully understandable that a 250W PV module will operate at lower power, depending on solar condition and temperature. However, when an engineer selects a converter for this PV installation, the maximum power should be considered. The authors did not add more explanations regarding the equations. A more comprehensive analysis is needed. The simpler gate driver circuits and the simple circuit construction should be compared with other existing topologies. A comparison with simulation results with other topologies showing voltages, currents, etc. would be very helpful.Author Response
Please see the attachment.

Reviewer 3 Report
On p.1, line 19 in the abstract section, this reviewer cannot find the “250-W, 72-cell PV module” set up throughout the manuscript.
Satisfied.On p.2, line 46, Please show a figure of the classical hard-switched converter with a note to compare the differences between two types of converters.
Satisfied.On p.2, line 48, at line 43 in p.2, you mentioned that raising the switching frequency results in increasing periodic switching and conduction losses. Then, why are you trying to use higher frequency more than a classical converter?
Satisfied.On p.2, line 54, please show related literature reviews (more than 5) regarding the soft-switching technique applied to PV modules and point out what are the main contributions compare to those.
Satisfied.On p.2, line 55, without a comparison table compared to classical converters, this insists cannot be made since there are no references.
Satisfied.Author Response
Authors would like to thank reviews for fruitful comments which enhance the quality of the paper.
Round 3
Reviewer 2 Report
The authors responded to all raised issues. The quality of the paper has been improved.
Author Response
Authors would like to thank reviews for fruitful comments which enhance the quality of the paper.